# Ontogenetic origins of cranial convergence between the extinct marsupial thylacine and placental gray wolf

Axel H. Newton [1,2,3✉], Vera Weisbecker [4], Andrew J. Pask[2,3,5] & Christy A. Hipsley [2,3,5✉]

Phenotypic convergence, describing the independent evolution of similar characteristics, offers unique insights into how natural selection influences developmental and molecular processes to generate shared adaptations. The extinct marsupial thylacine and placental gray wolf represent one of the most extraordinary cases of convergent evolution in mammals, sharing striking cranial similarities despite 160 million years of independent evolution. We digitally reconstructed their cranial ontogeny from birth to adulthood to examine how and when convergence arises through patterns of allometry, mosaicism, modularity, and integration. We find the thylacine and wolf crania develop along nearly parallel growth trajectories, despite lineage-specific constraints and heterochrony in timing of ossification. These constraints were found to enforce distinct cranial modularity and integration patterns during development, which were unable to explain their adult convergence. Instead, we identify a developmental origin for their convergent cranial morphologies through patterns of mosaic evolution, occurring within bone groups sharing conserved embryonic tissue origins. Interestingly, these patterns are accompanied by homoplasy in gene regulatory networks associated with neural crest cells, critical for skull patterning. Together, our findings establish empirical links between adaptive phenotypic and genotypic convergence and provides a digital resource for further investigations into the developmental basis of mammalian evolution.

[1] School of Biomedical Sciences, Monash University, Melbourne, VIC, Australia. [2] School of BioSciences, The University of Melbourne, Melbourne, VIC, Australia. [3] Department of Sciences, Museums Victoria, Melbourne, VIC, Australia. [4] College of Science and Engineering, Flinders University, Adelaide, SA, Australia. [5]These authors contributed equally: Andrew J. Pask, Christy A. Hipsley. ✉email: axel.newton@monash.edu; christy.hipsley@unimelb.edu.au

When the first European settlers arrived on the remote Australian island state of Tasmania, they were astonished to find a large, striped dog-like animal which, unlike other canids, had an abdominal pouch where it reared its young. Appropriately named *Thylacinus cynocephalus*, translating to "pouched-dog dog-headed"[1,2], the marsupial thylacine displayed remarkable similarities to placental canids[3–5], despite last sharing a common ancestor over 160 million years ago[6]. The thylacine and gray wolf (*Canis lupus*) are considered one of the most striking cases of convergent evolution in mammals, independently evolving nearly identical skull shapes[7] in response to shared carnivorous and predatory ecologies[8–11], despite differences in their post-cranial anatomy[11,12]. This example of convergent evolution offers an opportunity to determine how natural selection influences developmental and molecular processes to generate similar characteristics[13,14]. Comparative studies of mammalian ontogeny have provided important insights into differences underlying marsupial and placental development[15–18] and how developmental mode can impact early cranial ontogeny[19]. Building on these comparisons, we can explicitly test whether distantly related species with convergent morphologies have evolved homoplasy in developmental and molecular pathways to establish similarities in skull shape.

Development of the vertebrate skull is a deeply conserved process achieved through defined genetic cascades and cellular behaviors during early embryogenesis[20]. The cranial bones arise from three distinct embryological origins: the frontonasal process (FNP), first pharyngeal arch (PA), and paraxial mesoderm (MES). The FNP and PA are generated from independent streams of ectoderm-derived neural crest cells (NC) and form the bones of the anterior facial skeleton, whereas cells from the paraxial head mesoderm (MES) form the bones of the posterior neurocranium[20–22]. Each stream of cranial mesenchymal cells possesses their own intrinsic genetic programming, which directs specific patterns of cellular migration, proliferation, and ossification to form individual bone groups[20,23,24]. These cranial precursors form discrete developmental modules, where each cranial region can evolve and respond to selection independently, known as mosaicism (or modularity)[25]. However, modularity patterns can also shift throughout ontogeny[26]. During growth and maturation, bones arising from discrete developmental modules can group into larger, integrated (co-varied) traits[27] in response to changing selective pressures[28]. As such, modularity and integration are important drivers of phenotypic evolution, where modules with shared developmental or functional associations can be uniquely shaped by selection[25,29–31]. Evolutionary shifts in modularity have been shown to facilitate or constrain morphological variation[28,31,32], or when selection favors similar trait integration patterns among species, can promote the evolution of convergent phenotypes[3,26,33,34].

The homologous organization and modular hierarchy of the mammalian skull provides an ideal system for examining the influence of functional, developmental, and genetic associations underlying convergent phenotypes. Therian (marsupial and placental) mammals display a remarkably conserved pattern of six cranial modules, recovered from both fossil and extant species[35]. However, integration, or between-module covariation patterns, differ between marsupials and placentals, likely in response to their dichotomous modes of reproduction[36,37]. Placental mammals typically develop through an extended intra-uterine gestation and are born in a relatively advanced (precocial) state compared with marsupials. In contrast, marsupials are born after a short gestation in a hyper-altricial state (resembling a placental fetus[38]), after which they crawl into the mother's pouch to continue development through extended lactation and suckling[39]. The marsupial mode of reproduction requires accelerated morphogenesis and heterochronic gene expression in the forelimbs[40–42] and cranial

bones[16,43], which has been suggested to limit marsupial cranial disparity[38] and integration of the developing oral region[41]. However, the existence of marsupial "developmental constraint" has been challenged because marsupials have not always possessed limited diversity compared with placental mammals[44], and do not appear to exhibit adaptive constraints on forelimb[45,46] or skull shape variation[47]. Nevertheless, despite these differences, the remarkable convergence of cranial shape between adult thylacines and wolves suggest the independent evolution of similar underlying developmental processes.

Recent comparisons between the thylacine and wolf genomes[7] revealed extensive homoplasy in regulatory regions controlling craniofacial development[48], suggesting the evolution of shared molecular pathways underlying their convergence. In this study, we further examine the developmental processes that have led to the extraordinary cranial convergence between the extinct thylacine and gray wolf[3,7], building on a rare developmental series of thylacine pouch young[49]. By applying X-ray computed tomography (CT) to preserved museum specimens, we describe the cranial ontogeny of the thylacine and wolf from birth to adulthood, drawing comparisons with five extant marsupial species. Using 3D landmark and point cloud-based geometric morphometric analyses, we examine the onset and extent of cranial similarity in ontogenetic and allometric growth patterns to determine how these are correlated throughout development. We specifically examine whether there is greater similarity between cranial regions with conserved embryonic tissue origins, linking phenotypic convergence with genetic homoplasy in craniofacial tissues[48]. In addition, we examine module covariation patterns between the thylacine and wolf throughout key stages of development, to determine whether their adaptive cranial shapes arise through homoplasious modularity and integration patterns. Our findings offer novel insights into the mechanisms underlying mammalian evolution and provide new information on the cranial ontogeny of an extinct species.

## Results
**Ontogenetic allometry.** We sampled crania covering the complete developmental trajectories for the wolf, thylacine, and five additional extant marsupials, where specimens were available. These included the brush-tail possum (*Trichosurus vulpecula*), koala (*Phascolarctos cinereus*), woylie (*Bettongia penicillata*), Eastern quoll (*Dasyurus viverrinus*), and dunnart (*Sminthopsis sp.*). Crania were sampled to cover growth from earliest neonatal stages with nearly completely closed cranial sutures, to full-grown adults. The thylacine and wolf exhibit lineage-specific developmental patterns, exhibiting distinct heterochrony in their cranial ontogeny. Wolves are born after a gestation of ~65 days in a relatively altricial state but possess a largely ossified skull[50,51] (mean skull length 54.3 mm; $n = 4$, Supplementary data 1), with closure of all major cranial sutures except between the parietal and occipital, which remains partially open. At birth the wolf is substantially larger and more developed than the hyper-altricial thylacine, which by 1.5 weeks after its ~21–35 day gestation[52] had a total skull length of 12 mm ($n = 1$, Supplementary data 1) and retained open sutures between all major cranial bones[49]. By ~1.25-months-old, the thylacine showed equivalence in skull length (~31 mm) and development to a fetal wolf (36 mm; Fig. 1A) but did not appear morphologically similar to a newborn wolf until 2.5 months after birth (~75 days; skull length 58 mm; Fig. 1A, Supplementary data 1). As the 1.5-week-old thylacine cranium resembled an early embryonic state with open cranial sutures, we omitted it from further analyses.

Using landmark-based morphometrics of ontogenetic cranial shape (Fig. 1B), we examined cranial growth patterns between our sampled taxa. We found that each species displayed significant allometric growth, meaning that cranial shape was strongly

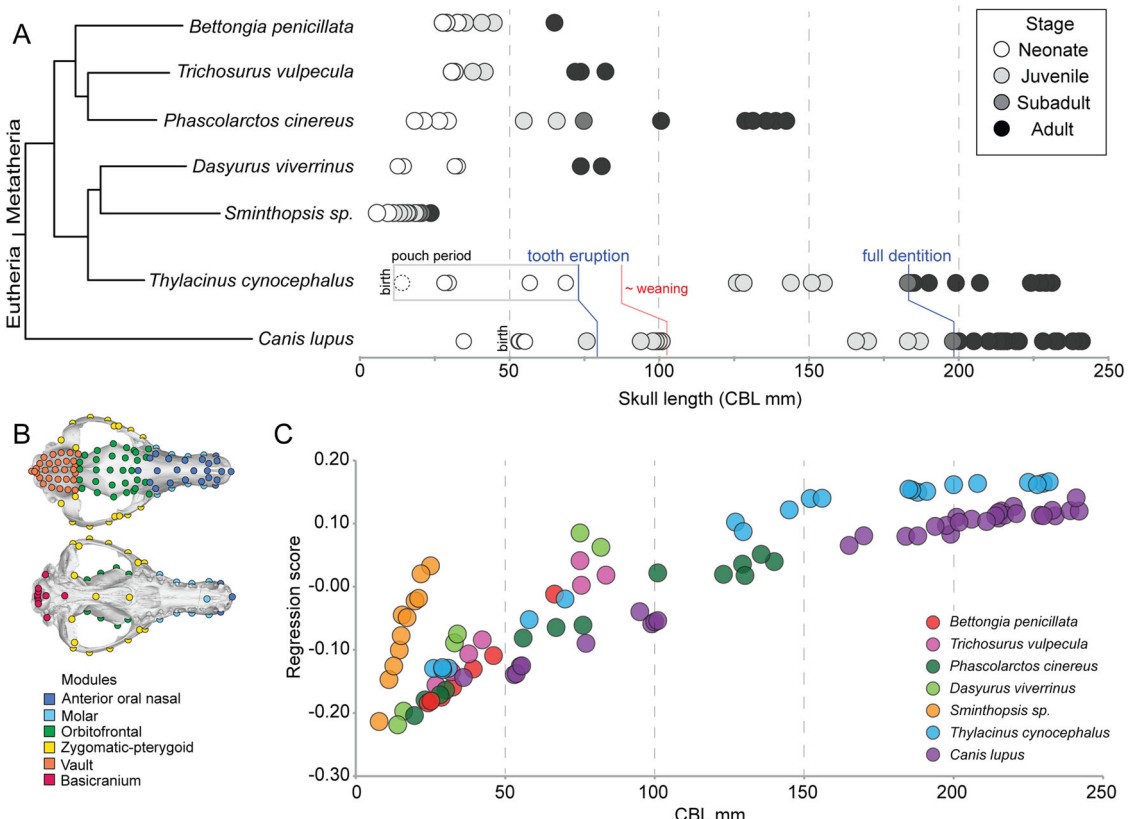

**Fig. 1 Taxa sampling, landmark and module locations, ontogenetic allometry. A** Crania were sampled from neonatal (white), juvenile (light gray), sub-adult (dark gray), and adult (black) individuals, from the marsupial thylacine (*Thylacinus cynocephalus*) and placental gray wolf (*Canis lupus*), and five additional marsupial species: the woylie (*Bettongia penicillata*), brush-tail possum (*Trichosurus vulpecula*), koala (*Phascolarctos cinereus*), Eastern quoll (*Dasyurus viverrinus*), and dunnart (*Sminthopsis sp.*). Crania were sampled to cover the complete developmental trajectory of each species. Skull lengths (CBL) for each sampled specimen are shown in millimetres. **B** Landmark locations (described in Supplementary data 2) used in the study, shown on an adult thylacine skull. Landmarks are colored by functional groups used in modularity analyses. **C** Ontogenetic allometry of sampled taxa revealed variation between developmental trajectories, though the thylacine and wolf displayed similar gradual patterns of shape change.

dependent on size (Table S1). The Procrustes analysis of variance (ANOVA) rejected the null hypothesis of common slopes ($F = 7.97$, $Z = 20.13$, $p = 0.001$), indicating that covariation of size and shape during postnatal ontogeny is not the same for all taxa. Allometric patterns were visualized as a scatter plot of individual regression scores, calculated as standardized shape scores from the regression of shape on size[53], against condylobasal length (CBL) (Fig. 1C) and centroid size (Fig S1), which were highly correlated ($r^2 = 0.99$; $p < 0.0001$; Fig S2). This revealed substantial variation among ontogenetic trajectories, with some species reaching maturity over a shorter range of skull lengths (e.g., *Sminthopsis sp.*). These smaller marsupials are characterized by tubular snouts, flat heads, smooth parietal bones (i.e., little to no sagittal crest) and a more bulbous cranium posteriorly than the larger-bodied carnivores. The thylacine and wolf display slightly more gradual shape change along the length of their trajectories (Fig. 1C), which cover roughly the same size range in our sample (thylacine CBL 29–232 mm, wolf CBL 36–242 mm). This is largely observed through parallel development of an elongated snout, widened zygomatic arches, and pronunciation of the sagittal crest.

Pairwise comparisons of slope vector lengths (magnitude) showed significant differences in the amount of shape change per unit size for 13 of the 21 species pairs (Table 1). The difference between thylacine and wolf was relatively small but significant ($p = 0.003$). In contrast, large comparative differences were observed between herbivorous and carnivorous taxa ($p \leq 0.003$ for each), such as *Phascolarctos* (koala) vs *Dasyurus* (quoll). Pairwise comparisons of slope vector angles, indicating the direction of

shape change per unit size, were significant for all but four species pairs, usually involving smaller taxa (i.e., *Sminthopsis*; Table 1). The largest angular differences, and hence lowest slope vector correlations ($r$), were between *Trichosurus*, *Phascolarctos,* and all other species (angular differences 35–48°, $p \leq 0.004$ for each), and *Thylacinus* vs *Bettongia* (37°, $p = 0.005$). The slope vector correlation between thylacine and wolf was close to 1 ($r = 0.91$, $p = 0.001$), indicating nearly parallel slopes. The only stronger angular correlations in the data set were between closely related carnivorous dasyuromorphs, *Dasyurus* vs *Thylacinus,* and *Sminthopsis* vs *Dasyurus*.

**Cranial disparity and convergence.** We next measured ontogenetic variation in cranial shape using principal component analysis (PCA). Over three quarters (77%) of the total shape variation was contained in the first three axes, with 54% explained by PC1 alone. PC1 describes the development from a bulbous and short-faced neonate (PC1−) to a mature adult skull shape with narrowing of the brain case and elongation of the snout (PC1+ ; Fig. 2A). PC1 was strongly positively correlated with log-transformed skull length (logCBL: CBL; Pearson's correlation $r = 0.84$, $p < 0.00001$) suggesting it is reasonable proxy for age-related changes in cranial shape[50,54]. PC2 (15%) separated taxa based on feeding ecology, describing widening of the cranial vault and zygomatic arches, as well as blunting of the snout (Fig. 2A, Fig S3). Carnivorous taxa and the primarily fungivorous *Bettongia* exhibited parallel growth trajectories, whereas herbivorous taxa displayed divergent patterns of

**Table 1 Ontogenetic allometry — differences in slope vector length and angle between sampled species pairs.**

| Species pair | Slope vector length | P value | Slope vector correlation (r) | Slope vector angle (degrees) | P value |
|---|---|---|---|---|---|
| Bettongia–Canis | 0.0341 | 0.174 | 0.834 | 33.467 | **0.026** |
| Bettongia–Dasyurus | 0.0345 | 0.172 | 0.853 | 31.428 | 0.125 |
| Bettongia–Phascolarctos | 0.0533 | **0.038** | 0.697 | 45.828 | **0.001** |
| Bettongia–Sminthopsis | 0.0350 | 0.186 | 0.819 | 34.998 | 0.147 |
| Bettongia–Thylacinus | 0.0606 | **0.019** | 0.796 | 37.209 | **0.005** |
| Bettongia–Trichosurus | 0.0275 | 0.325 | 0.673 | 47.697 | **0.004** |
| Canis–Dasyurus | 0.0004 | 0.973 | 0.903 | 25.429 | **0.002** |
| Canis–Phascolarctos | 0.0192 | **0.042** | 0.817 | 35.206 | **0.001** |
| Canis–Sminthopsis | 0.0691 | **0.001** | 0.845 | 32.317 | **0.006** |
| **Canis–Thylacinus** | 0.0265 | **0.003** | 0.911 | 24.402 | **0.001** |
| Canis–Trichosurus | 0.0616 | **0.001** | 0.742 | 42.124 | **0.001** |
| Dasyurus–Phascolarctos | 0.0189 | 0.177 | 0.806 | 36.269 | **0.001** |
| Dasyurus–Sminthopsis | 0.0695 | **0.001** | 0.927 | 21.982 | 0.536 |
| Dasyurus–Thylacinus | 0.0261 | **0.048** | 0.917 | 23.452 | **0.003** |
| Dasyurus–Trichosurus | 0.0619 | **0.003** | 0.764 | 40.155 | **0.001** |
| Phascolarctos–Sminthopsis | 0.0883 | **0.001** | 0.732 | 42.938 | **0.001** |
| Phascolarctos–Thylacinus | 0.0072 | 0.431 | 0.816 | 35.277 | **0.001** |
| Phascolarctos–Trichosurus | 0.0808 | **0.001** | 0.779 | 38.857 | **0.001** |
| Sminthopsis–Thylacinus | 0.0956 | **0.001** | 0.894 | 26.557 | 0.055 |
| Sminthopsis–Trichosurus | 0.0075 | 0.744 | 0.760 | 40.568 | **0.009** |
| Thylacinus–Trichosurus | 0.0880 | **0.001** | 0.797 | 37.123 | **0.001** |

Significant P values are in bold.

development (Fig. 2A, Fig S3). PC3 (8% of variation) separated the wolf from the carnivorous marsupials, reflecting its robust skull including widening of the molar row and pronunciation of the interparietal process (Fig S3).

Given the dichotomous life histories and reproductive strategies of marsupials and placental mammals[15,39], we hypothesized that the thylacine and wolf would display disparate early cranial shapes, only to converge later in their juvenile to adult stages of development. Instead, the thylacine and wolf displayed parallel, nearly overlapping, patterns of PC1–2 shape similarity throughout ontogeny (Fig. 2a), reflecting their allometric growth patterns (Fig. 1c, Table 1). Furthermore, the thylacine displayed greater ontogenetic similarities to the wolf than to any of its close marsupial relatives (Fig. 2a). The fetal wolf (36 mm) nearly overlapped in cranial shape with the 1-month-old thylacine (31 mm), indicating both species established similar skull morphologies (i.e., a bulbous cranium with a short snout) at the earliest stages of ontogeny, despite their developmental heterochrony. These similarities persisted through to their adult cranial shape[3,7], as seen by the homoplasious development of a long, narrow snout and dorsally expanded skull roof forming a pronounced sagittal crest.

**Convergence in cranial regions with shared embryological origins.** To investigate the extent to which genetic convergence between the thylacine and wolf may be reflected in cranial growth patterns, we examined morphological variation in cranial sub-regions arising from discrete embryological tissue origins. Cranial landmarks were subsampled into three datasets[55] with landmarks covering cranial regions arising from distinct tissue origins—bones of the NC-derived FNP (Fig. 3C) or first PA (Fig. 3D), or bones of paraxial head MES (Fig. 3E)[20]. Each subsampled dataset reflected similar patterns observed in the complete morphospace, with over three quarters of the total shape variance held in PC1–3 (FNP = 80%; PA = 80%; MES = 72%). However, we detected strong ontogenetic similarity and adult convergence in thylacine and wolf bones arising from FNP NC (i.e., frontal, nasal, and premaxilla bones; Fig. 2B) and mesodermal origins (i.e., parietal and occipital bones; Fig. 2D). This similarity was not observed in bones of PA origin (i.e., zygomatic, maxilla, lacrimal, temporal,

sphenoid, palatine), which retained distinct shapes between the thylacine and wolf throughout ontogeny (Fig. 2C). Together, these data suggest that cranial convergence may be mosaic, with homoplasy arising specifically within FNP-NC and MES developmental modules[25].

**Point cloud visualization of cranial convergence.** To further visualize regions of the crania that displayed the greatest variation during ontogeny, we generated point cloud comparisons of similar sized and stage-matched thylacine and wolf skulls. Regions with small cloud-to-cloud distances are shown as cool colors (blue and green) while large distances are expressed as warm colors (red and purple). Stage-matched comparisons were found to complement the ontogenetic cranial morphospace (Fig. 2), where the thylacine and wolf crania show strong similarity during the neonatal, juvenile, and adult stages of ontogeny (Fig. 3). The largest overall differences were observed in the basicranium, particularly the occipital condyles, basioccipital, and auditory bulla (Fig. 3A, B). During maturation, shape disparity arose around the nasal aperture and braincase, with larger differences occurring in the positioning of the molar row and pronunciation of the interparietal process (Fig. 3C). However, these adult cranial differences were relatively minor, highlighted by similar-scaled comparisons with the thylacine's close relative, the Eastern quoll (*Dasyurus viverrinus*) (Fig. 3D), showing overall dissimilarity particularly in the size and shape of the anterior rostrum.

**Cranial modularity and integration.** Finally, we investigated ontogenetic patterns of cranial modularity and integration (covariation). We compared thylacine and wolf module covariation patterns using covariance ratio analysis (CR)[56] of the three embryonic tissue origins (FNP-NC, PA-NC, MES) and an established, mammalian six functional module model[35]. Pairwise CR comparisons between each cranial module in the neonatal and juvenile thylacine and wolf stages were similarly large (CR > 0.85), suggesting all cranial functional modules remain integrated during the early patterning and remodelling of the cranial bones (Fig. 4, Table S2). However, these differed markedly in adults (Fig. 4, Table S2). Adult wolves possess low CR values (CR ≤ 0.85) for

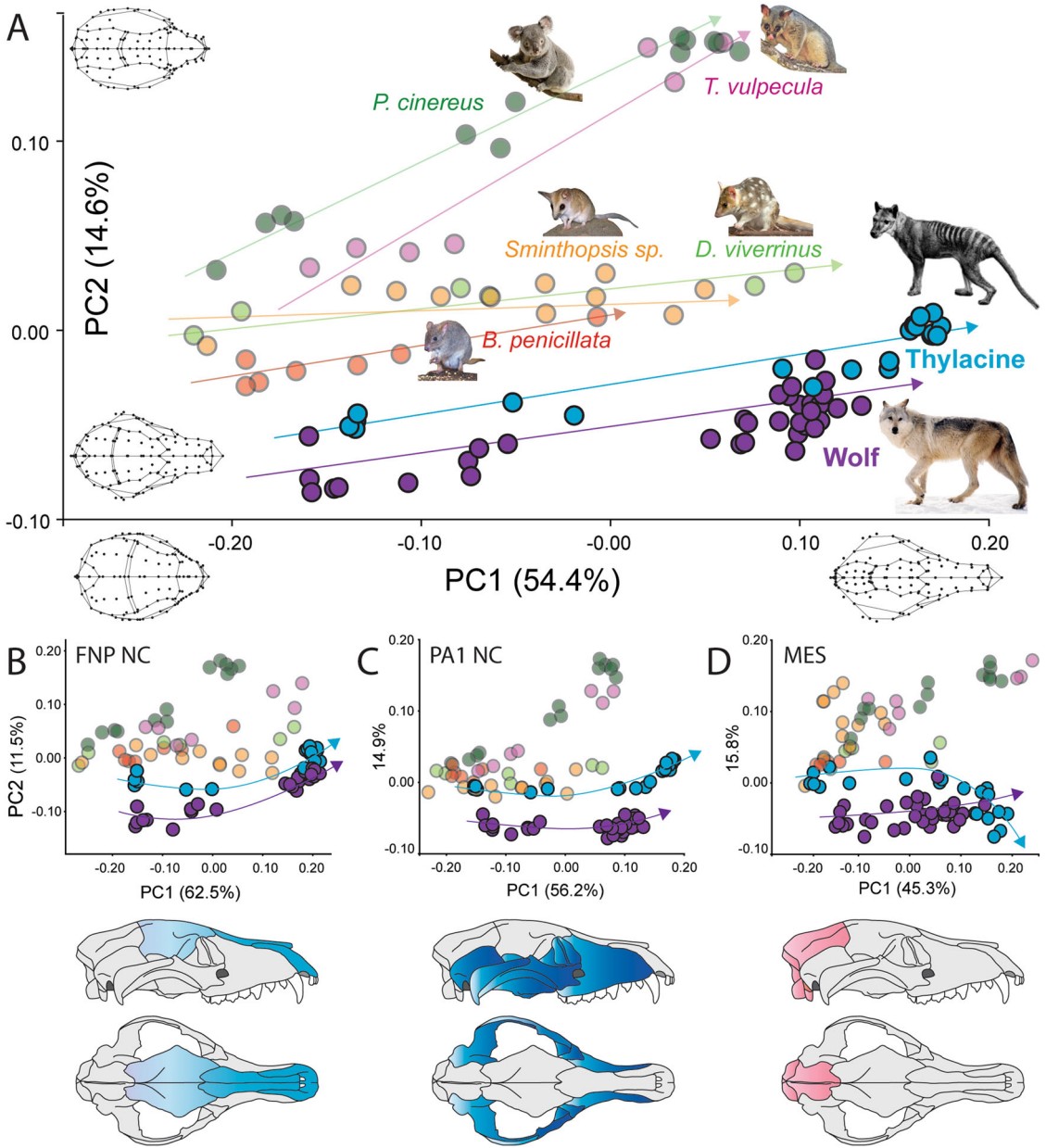

**Fig. 2 Ontogenetic cranial variation and mosaic evolution. A** Principal component analysis of ontogenetic cranial shape for each species included in the study. PC1 represents age-related shape change (left to right), whereas PC2 separates herbivorous and carnivorous taxa. The thylacine and wolf display parallel similarities throughout ontogeny, compared with other marsupials. **B–D** Subsampling of cranial shape into bone groups with shared embryonic tissue origins. The thylacine and wolf show shape overlap between bones of **B** FNP and **D** MES origin, but not in bones of **C** PA origin. Animal images were used under CC BY 4.0 open licence.

each cranial module comparison, suggesting the adult wolf skull becomes more modular during maturation[55,57]. In contrast, adult thylacines returned low CR values for cranial regions associated with the neurocranium (basicranium & cranial vault; CR < 0.85) but expressed large values (CR > 0.85) between the facial regions (orbitofrontal, oral-nasal, molar, zygomatic-pterygoid; FNP, PA). Together this suggests that while the thylacine neurocranium becomes more modular, the facial bones remain highly integrated throughout ontogeny[55,57]. CR analysis of the allometry-corrected residuals yielded largely similar modularization of the crania across development, but with slightly greater modularity (CR < 0.8) between four partitions at the juvenile stage (orbitofrontal–cranial vault and cranial vault-basicranium in the thylacine, molar-cranial vault, and cranial vault-basicranium in the wolf; Table S3).

## Discussion

The extinct thylacine and gray wolf present an exceptional model of convergent evolution, with recent studies supporting extreme phenotypic similarity in the skull[7,9] and accompanying genomic homoplasy in craniofacial gene-regulatory elements[48]. However, the underlying developmental processes that link these homoplasies are not well understood. Using detailed taxonomic, landmark, and point cloud-based comparisons, we demonstrate that the thylacine displays a departure from the tightly constrained marsupial pattern[3,7,58], instead sharing parallel similarities in cranial development (Figs. 2, 3) and allometric growth patterns (Table 1) with the wolf, instead of its close marsupial relatives, across its entire postnatal ontogeny. The inclusion of additional taxa representing broader sources of cranial variation (i.e., placental herbivores and carnivores), or larger-bodied carnivorous marsupials such as the

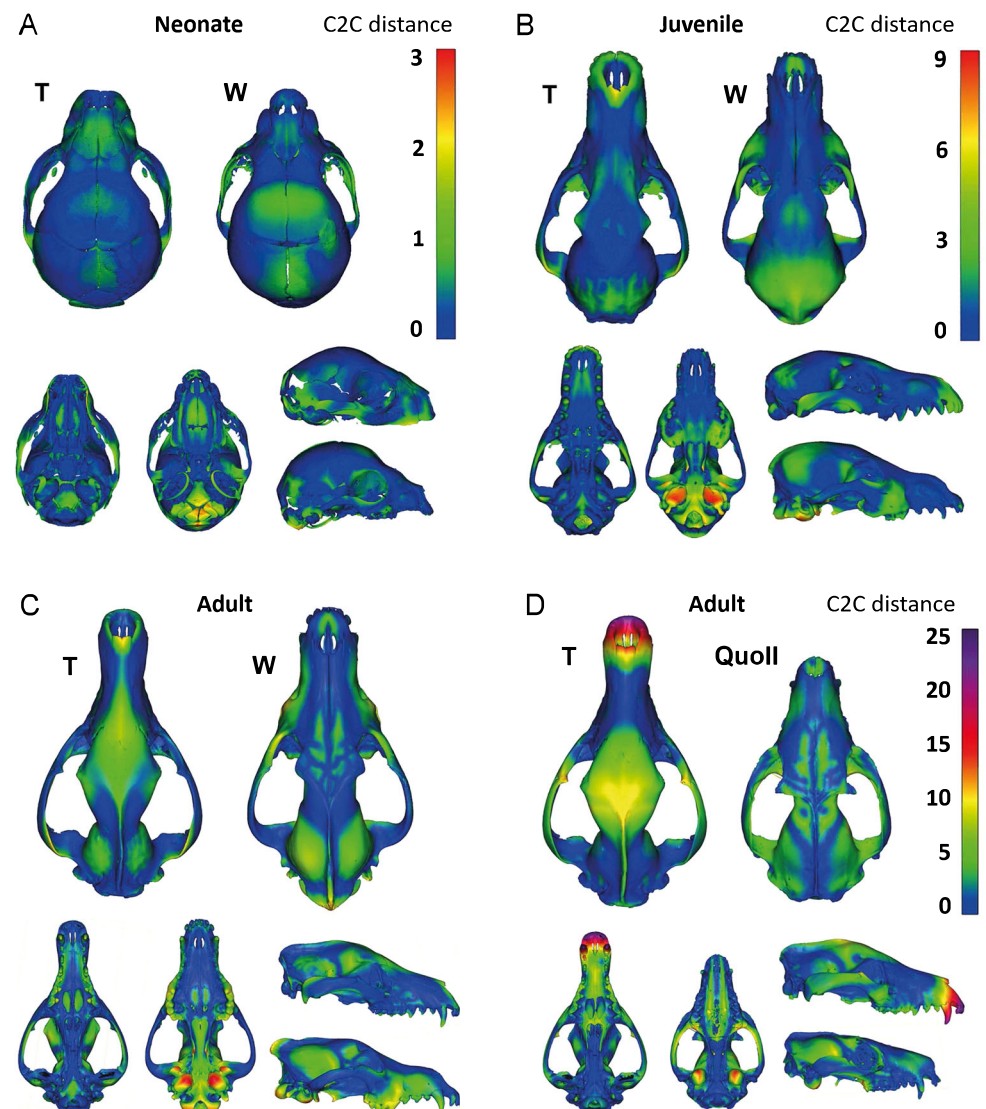

**Fig. 3 Point cloud comparisons of thylacine and wolf ontogenetic cranial shape.** Small point-to-point differences are expressed as cool colors (blue/green) while large differences are observed as warm colors (red/purple). Warmer colors denote greater Euclidean distances between point clouds. Minor shape differences are observed between thylacine (T) and wolf (W) crania at **A** neonatal, **B** juvenile, and **C** adult stages. Similarities become apparent when the thylacine is compared with its extant relative, **d** the Eastern quoll.

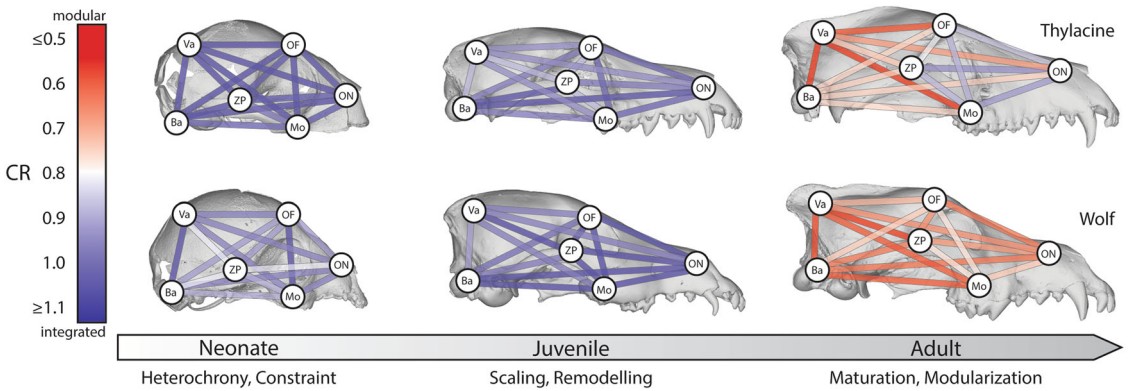

**Fig. 4 Network diagrams of module covariation patterns during ontogeny.** Pairwise covariance ratio (CR) coefficients between cranial landmark modules (see Table S2) for thylacine and wolf on neonate, juvenile, and adult 3D rendered skulls. Low CR values are expressed by warm colors, indicating modularity, whereas cool colors represent high CR values, suggesting integration. Neonatal and juvenile modules show high CR ratios (integration). The adult thylacine retains an integrated facial skeleton, while the adult wolf skull becomes modular. *Ba* basicranium, *Mo* molar, *OF* orbitofrontal, *ON* oral-nasal, *Va* vault, *ZP* zygomatic-pterygoid.

Tasmanian devil—*Sarcophilus harrisii*, or tiger quoll—*Dasyurus maculatus*[3,5,58], may reveal more subtle homoplasious growth patterns. However, the broad similarities observed between the thylacine and wolf are nonetheless remarkable given their 160 million year divergence[6], dichotomous reproductive strategies, and observed developmental heterochrony (Fig. 1)[15,36,58,59].

The parallel development of the thylacine and wolf crania complement post-cranial skeletal growth patterns in which the thylacine also develops with greater similarity to other large placental carnivores than to marsupials[49], irrespective of the differences in their forelimb anatomy[12,60]. These similar developmental trajectories are likely an adaptive response to shared ecological niches and biomechanical demands of predation, i.e., the development of robust cranial bones and musculature to generate high bite force quotients[8–11]. Importantly, the thylacine appears to have evolved unique developmental mechanisms to facilitate its adaptive convergence, making this comparison an extraordinary evolutionary model to scrutinize the developmental processes underlying convergent evolution.

To better understand the ontogenetic origins of the thylacine-wolf convergence, we subsampled cranial shape data into bone groups arising from discrete embryological tissues. This revealed disparate patterns of ontogenetic shape between the FNP, PA, and MES groups, suggesting they are mosaic and able to evolve and adapt semi-independently[25]. PA-FNP and MES bones converged between the thylacine and wolf, whereas bones of PA-NC origin were distinct, instead showing signatures of constrained shape between marsupials (Fig. 2). PA-NC cells migrate and ossify to establish the bones of the masticatory (oral) apparatus, which in marsupials, is accelerated to facilitate the functional demands of altricial suckling[36,59,61,62]. These constraints may produce stabilizing selection and reduced shape evolvability of the oral bones within marsupial taxa[34], thus restricting homoplasy with placental mammals. In contrast, the observed convergence of FNP-NC and MES bone groups between thylacine and wolf suggests their similarities stem from a common developmental origin. Here, adaptive molecular evolution may arise within FNP-NC and MES embryonic cell lineages, regulating the convergent development of these bone groups. This hypothesis is supported by the recent identification of thylacine-wolf homoplasy in gene-regulatory elements of major patterning genes and developmental pathways regulating cranial mesenchyme migration, differentiation, and ossification[48]. This is further supported where perturbations to patterning genes within cranial NC cells can directly alter facial morphology[24]. Taken together, these observations establish tangible links between phenotypic, developmental, and genomic convergence. That is, homoplasy targeting gene-regulatory networks within cells of FNP-NC or MES origin may drive the development of similar, adaptive cranial shapes throughout ontogeny.

In addition to cranial mosaicism, homoplasious adaptations may arise through selection favoring shared module covariation or integration patterns[3,26,33,34]. Surprisingly though, we found that the thylacine and wolf exhibit distinct patterns of functional modularity and integration during their development. Although the wolf showed overall increases in cranial modularity during postnatal ontogeny[63], the thylacine retained high covariation between facial modules, particularly between bones of PA-NC origin, further reflecting its marsupial biology (Fig. 4)[36,49,61]. As such, the thylacine and wolf have not evolved their striking cranial convergence through homoplasious functional integration patterns, owing to their lineage-specific constraints[64]. Placental mammals display an incredible range of cranial variation[58,59] owing to the relaxed constraints, and increased modularity, associated with their comparatively extended gestation[65]. In comparison, marsupials display reduced overall cranial variability and plasticity[58], suggested to occur in response to the developmental and functional constraints associated with their mode of reproduction[15,37,58,64]. Although

previous vertebrate studies have shown that selection can favor modularity and integration patterns to promote morphological convergence between closely related taxa[33,66,67], our results demonstrate that convergent phenotypes may not always be generated through homoplasious patterns of functional integration, instead arising through alternate developmental pathways.

The ontogenetic thylacine-wolf comparisons presented in this study provide unique and novel insights into the developmental processes underlying morphological convergence and evolution in mammals. Although mammalian lineage-specific constraints influence embryonic and neonatal stages of development, our data suggest that these constraints can be rapidly overcome, likely as an adaptive response to shared ecological niches[9,10]. We show that the cranial similarities between the thylacine and wolf are in fact mosaic[25], where cranial bone groups with strong shape similarity arise from conserved embryological origins. Strikingly, these patterns are accompanied by genomic homoplasy within these key embryonic cell populations[48], providing strong empirical links between phenotypic and genotypic homoplasy. These findings prompt exciting new research avenues to examine how molecular changes within tissue-specific regulatory networks may influence craniofacial shape disparity and convergence. The identification of these developmental similarities and differences provides a novel framework for defining the causative factors underpinning skull evolution and the remarkable cranial convergence seen between the thylacine and wolf[7].

## Methods

**Sampling**. Three-dimensional digital cranial models for seven species of mammals, including the thylacine and gray wolf, were acquired from published studies[7,49–51,62,68], public repositories (Digimorph, MorphoSource, Digital Morphology Museum), or newly generated using X-ray CT on museum specimens (Supplementary data 1). Five marsupial species were chosen for comparison with the thylacine, subject to their availability. We sampled crania from the dunnart (*Sminthopsis sp.*) and eastern quoll (*Dasyurus viverrinus*), which belong to the same carnivorous order as the thylacine (Dasyuromorphia) and show similarities in adult cranial morphologies (e.g., long and narrow snouts)[7], but have different body sizes and life histories. We were unable, however, to sample the larger-bodied Tasmanian devil (*Sarcophilus harrisii*) or tiger quoll (*Dasyurus maculatus*) owing to a lack of cataloged postnatal specimens. We also included divergent taxa belonging to the distantly related order Diprotodontia, which display disparate cranial morphologies[7], owing to a mainly herbivorous diet (koala, *Phascolarctos cinereus*; brush-tail possum, *Trichosurus xvulpecula*; and woylie, *Bettongia penicillata*), and unique ecologies (e.g., terrestrial bipedal or arboreal). CT scanning was performed at the School of Earth Sciences, University of Melbourne, in a GE Phoenix Nanotom M and 3D volumes were reconstructed in datos|x-reconstruction software (GE Sensing & Inspection Technologies GmbH, Wunstorf, Germany). Crania were isolated from the skeleton in VGStudio Max 3.0 (Volume Graphics, Heidelberg, Germany) and exported as surface meshes. The final data set consisted of 101 individuals spanning the postnatal growth period of each species (6–35 individuals/species, average = 14; Fig. 1A). Age class was determined by tooth eruption patterns and life history, where available. Generally, individuals without any erupted teeth were considered neonates; individuals with at least one erupted tooth were considered juveniles; individuals with all but one of their adult dentition were considered subadults and individuals with all of their teeth were considered adults[69,70] (Supplementary data 1).

**Geometric morphometric analyses**. Biological shape was captured by 128 landmarks placed across the cranial surface in Landmark Editor (Institute of Data Analysis and Visualization, UC Davis, USA), including 30 anatomical landmarks from previous studies[3,7] and 98 additional landmarks in the form of points, patches and sliding semi-landmarks (Fig. 1B; Supplementary data 2). The final landmark data set can be found in Supplementary data 3–5, and associated cranial CT data and surface meshes are publicly available on MorphoSource (www.morphosource.org, project number P1124). Geometric information was extracted from the landmark co-ordinates by a generalized Procrustes fit in the R package geomorph v3.1.2[71]. The resulting Procrustes co-ordinates, representing the symmetric component of shape variation after translating, scaling, and rotating all individuals to a common centroid, were used as shape variables in all analyses. CBL, measured from the anterior-most tip of the snout to the posterior surface of the occipital condyles, was used to determine relative cranial size across development (Supplementary data 1). CBL showed a strong linear relationship with centroid size in the data set (Fig S2) so was used in subsequent analyses. Species ontogenetic allometries were assessed by regression of cranial shape on log-transformed CBL and compared using Procrustes ANOVA. Given a significant species*size interaction, species pairwise comparisons were performed to identify differences in allometric vectors (degree of shape change), including slope vector

length, correlation and angle (in degrees). This was achieved using the pairwise function on the unique species allometry model in the R package RRPP v0.6.2[72], which compares least-squares (LS) means while accounting for allometry and species effects.

To assess the role of modularity in morphological convergence of thylacine and wolf, landmarks were partitioned according to two model structures. The embryonic tissue origin model included three modules based on their derivation from cranial/pharyngeal arch neural crest, or paraxial head mesodermal origin (1—FNP and 2—first PA NC; and 3—head MES). The functional module hypothesis included six functional groups supported by previous studies on mammals[35,73], including four of the species examined here. These modules were slightly altered to match our landmark placements, so that each module was comprised of sets of individual bones, rather than parts of bones, which corresponded to tissue origin(s), i.e., FNP-NC-derived frontals, nasals, and premaxilla; PA NC-derived maxilla, zygomatic arch, palate, sphenoid, pterygoid, and other small bones; and MES-derived parietals and occipitals forming part of the neurocranium and cranial base. The resulting modules were 1—oral-nasal (FNP/PA), 2—molar (PA), 3—zygomatic-pterygoid (PA), 4—orbitofrontal (FNP), 5—cranial vault (MES), and 6—basicranium (MES).

Changes in the degree of cranial modularity during postnatal development were identified using covariance ratio analysis (CR)[56], describing the covariation of landmarks between modules relative to the covariation within them. CR coefficients range from zero to positive values, with values between zero and one indicating greater covariation within than between modules (i.e., greater modular structure), whereas values larger than one indicate greater covariation between, than within, modules (i.e., less modular–more integrated structure). A CR coefficient of one is expected for random sets of landmarks, since levels of covariation between and within modules should be, on average, the same[56]. CR was estimated separately for the neonate, juvenile, and adult stages of the thylacine and wolf for the full landmark data set and after correcting for allometric effects, using the residuals from the species-specific Procrustes ANOVA. Given the low specimen-to-landmark ratio in our data set, we also performed a random data simulation for similar module partition sizes at our smallest developmental stage sample ($N = 6$), which returned reasonable (0.057) type I error rates.

Cranial shape changes during ontogeny were illustrated by wireframe graphs showing shifts in landmark positions. For visualization of cranial shape differences between thylacine and wolf at distinct developmental stages (neonate, juvenile, adult), specimens with similar CBLs were selected for each species and aligned using CloudCompare v2.10.2 (www.cloudcompare.org/). CloudCompare differs from landmark-based analyses in that it renders the entire cranial mesh as a dense cloud of equidistantly spaced points. Point clouds for each stage were aligned using the "Align (point pairs picking)" tool with eight equivalent points on each skull (considered a rough registration), followed by the "Fine registration" tool, both with thylacine as the reference model and adjust scale selected. Meshes were subsampled to 10 million points and compared using "Compute cloud/cloud or cloud/mesh distance", first with thylacine as the reference and then with wolf. Resulting Euclidean distances between skulls were displayed in color scale, reflecting the nearest neighbour distance for each point of the compared clouds.

**Statistics and reproducibility**. All statistical tests were performed in MorphoJ and the R packages geomorph v3.1.2[71] and RRPP v0.4.2.9[72], with $P$ values generated from a randomized residual permutation procedure of 10,000 iterations. No special ethical considerations were necessary for this study. Taxon sampling was performed in accordance with relevant guidelines and regulations.

**Reporting summary**. Further information on research design is available in the Nature Research Reporting Summary linked to this article.

## Data availability

Detailed specimen information and source is provided in Supplementary Data 1. Raw landmark co-ordinates with associated classifier and covariate data is included in Supplementary Data files 3–5. CT data and cranial models generated for this study are publicly available on MorphoSource (www.morphosource.org; project number P1124).

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

## Acknowledgements

We thank Aren Gunderson and Link Olson from the University of Alaska Museum for loan of wolf specimens, Katie Smith and Kevin Rowe from Museums Victoria for loan of thylacine specimens; Justin Adams, Allister Evans, and Tahlia Pollock (Monash University) for access to juvenile thylacine CT scan data and assistance with specimen aging based on tooth eruption patterns; Stephan Spiekman for access to marsupial pouch young cranial models; Sharleen Sakai (National Science Foundation; ISO 1146614) for access to juvenile and adult wolf cranial models; and Jay Black from the Trace Analysis for Chemical, Earth and Environmental Sciences (TrACEES) platform from the Melbourne Collaborative Infrastructure Research Program at the University of Melbourne for CT scanning.

## Author contributions

A.H.N., C.A.H., and A.J.P. conceived the study. V.W. assisted with experimental design. A.H.N. performed CT data acquisition and reconstruction, meshing, and landmark acquisition. A.H.N. and C.A.H. performed morphometric analysis. C.A.H. performed statistical testing and modularity analysis. A.H.N. wrote the manuscript with assistance from all co-authors.

## Competing interests

The authors declare no competing interests.
