## [Peer Review File · Communications Biology]

Reviewers' Comments:

Reviewer #1:

Remarks to the Author:

Copied in by editor

Review COMMSBIO-20-2034-T

General comments: This is a study about an interesting topic, the cranial convergence between mammals (a marsupial and a placental), approached with a suitable technique such as geometric morphometric. The structure is clear and it is well written. The graphics generated are very good. In general, materials and methods used are appropriate to the aim of the authors, although I found only one thing more complex to solve:

Cranial similarities and/or differences are directly related to the species that we choose to compare. When a study includes species, which are morphologically extreme or have a very different diet (too small or with a completely different diet or prey size):

Bettongia penicillata (1.8 kg, Mycophagous),

Dasyurus viverrinus (1.9 kg, eats insects and small vertebrates),

Phascolarctos cinereus (11 kg, folivore),

Sminthopsis crassicaudata (20 g, eats invertebrates),

S. macroura (25 g, eats mostly invertebrates),

Trichosurus vulpecula (4 kg, Mostly folivore)

these species gathered the greatest amount of variation, blurring more subtle patterns that may appear in the remaining species. In fact, this forces the similitude between *Thylacinus cynocephalus* and *Canis lupus* because they are more different. It would be more interesting perform the analysis with *Dasyurus maculatus* (1.5-5 kg, eats small vertebrates), *Sarcophilus harrissi* (8-14 kg, hypercarnivorous, eats medium vertebrates), and *Thylacinus cynocephalus* (29.5 kg, eats smaller prey relative to its size), and to compare them with *Canis lupus* (60 kg, hypercarnivorous). These three species belong to the carnivore guild of large mammals (Marsupials), and they are used in many works that report the convergence between *Thylacinus cynocephalus* and canids.

If the authors change the sample by using three carnivorous medium-sized mammals (e.g. *Dasyurus maculatus*, *Sarcophilus harrissi*, and *Thylacinus cynocephalus*),

I think that the pattern observed will be the same, but they will see the more subtle differences between *T. cynocephalus* and *Canis lupus* directly in a PC analysis without necessity to use cloud compare or other software to see that.

I think that it would be a better comparison and reduce the analyses.

Introduction

1. Line 43-45 "The thylacine and grey wolf (*Canis lupus*) are one of the most striking cases of convergent evolution in mammals, independently evolving nearly identical skull shapes ..."

What about previous papers cited here that show more relationship between *Thylacinus cynocephalus* and *Vulpes vulpes* than *Canis lupus*?

2. Line 73-75 "Placental mammals typically develop through an extended intrauterine gestation and are generally born in an advanced (precocial) state."

In the case of wolves, this is not accurate because they are altricial. Wolves are born deaf, blind and helpless. They are unable to move and even to regulate their own body temperature.

3. The introduction would beneficiate from a brief paragraph where authors explain why is interesting the study of the comparative ontogeny. In addition, it would be a great moment to cite previous works in carnivores and marsupials ontogeny.

Results

4. Line 112 Skull length 58.8 mm (mean)
5. Line 114 more developed in which way? In relation to shape, to size? Please describe
6. Line 115 Skull length 37.1 mm (mean)
7. Line 118 similar in which sense? Please describe
8. Line 111-120. Please modify the paragraph
9. Line 126-127 What CAC means? Please explain this in material and methods section
10. Line 130-134 I would like a description about shape changes during ontogeny focused on the two species with phenotypic convergence instead of the other marsupials.
11. Line 135 Please explain how the "Pairwise comparisons of slope vector lengths" was performed (in material and methods).
12. Line 150 In figure 2 the PC1 explained 54.4%

Discussion

13. Line 223 Please, try to integrate to the discussion the previous papers that compare the two same species.
14. Line 234-236 "These similarities are likely an adaptive response to the biomechanical demands of predation and shared ecological niches."
The discussion would benefit from a deeper description of the relation to the cranial shape and the biomechanical demands and predatory behavior that authors propose that the two compared species share.

Materials & Methods

15. Line 296-297 I cannot find digital cranial models neither in the paper nor in supplementary material for Ramírez-Chaves et al. 2016 (cited in Supplementary 1) Do materials acquire from published studies or from personal communication? Please modify if it is necessary.
 16. Line 309-310 "...with age class determined by tooth eruption and life history data, when available."
Please, describe more the determination of age classes, maybe a table would be fine.
 17. Line 315 Table S2 is not about landmarks
 18. Line 320-322 Why is necessary to take measures as CBL for comparisons if you have centroid size? You could perform regressions between log Centroid size and Procrustes coordinates.
 19. Line 346 more integrated
- ## Minor details
20. If it is possible, separate the figure 1 en two figures. The first of them should be the figure 1B.
 21. Please add captions to supplementary material.

Reviewer #2:

Remarks to the Author:

Newton and colleagues present a study of ontogenetic changes in the cranial morphology of the thylacine and the wolf, with additional analyses of those trajectories in the context of bone developmental and functional modules. They found broad convergence in the patterns of overall cranial shape changes between the two species, together with differences in the shifting modularity of within-cranium bone modules. This complex pattern of convergence and constraint is attributed in part to fundamental differences in the life histories of the two predators.

The manuscript is well-written, and the figures and tables are very informative for interpreting the authors' findings. This is a great study that adds clarity to the claim of morphological and

ecological convergence between the thylacine and the wolf, through quantitative analyses of cranial shape changes through ontogeny. The statistical methods appear to be robust and carefully chosen. The comparative nature of such morphological analyses typically would require phylogenetic comparative methods to account for phylogenetic statistical non-independence; in the case of this study, the divergence times between thylacine and wolf are so deep, any effect of phylogenetic relatedness in the metatherian part of the dataset most likely would not have altered the conclusions presented by the authors. The figures add greatly to the interesting results reported.

I have just two suggestions for improvement, both concerning data reporting:

-P17L310: according to Table S1, the CT images used in this study are a combination of openly accessible and not openly accessible data. For maximum transparency, please provide explanations for the different accessibility levels of the raw data used in the analyses. Neonatal and juvenile specimens of any wild mammal species are uncommon in museum collections, and especially so for an extinct taxon such as the thylacine. The importance of this dataset cannot be overstated, for future work that build on the current study, as well as other lines of research on the ontogeny and evolution of the thylacine. Therefore, increasing data accessibility could be one of the significant contributions presented by this study.

-P20L370: surely the analyses were done with the usual ethical standards in mind (Data free of manipulation, digital datasets used with permission, etc.)? I suggest "No special ethical considerations were necessary in this study" instead.

Reviewer #3:

Remarks to the Author:

The manuscript entitled "Ontogenetic origins of cranial convergence between the extinct marsupial thylacine and placental grey wolf" aim to find similarities and differences in the ontogenetic growth patterns between the thylacine and the wolf, two species that represent one of the most iconic examples of convergence between marsupials and placentals. Their starting hypothesis is that some similarities should be found, but also some differences due to the characteristics developmental constraints of each group. They also explore patterns of modularity and integration through ontogeny, which are very important aspects that influence morphological change. In my opinion, the hypotheses are clear, the analyses are correct and their conclusions are supported by the results. I strongly recommend the publication of this article in *Communications Biology*. I just have some suggestions for improving the manuscript that the authors may find useful. They are detailed below:

Page 3, line 45. There are also evidences of a lack of convergence in their predatory behavior or hunting strategy, as shown by Figueirido and Janis (2011) and Janis and Figueirido (2014).

Page 3, line 52. In my opinion, this sentence is not clear enough about which are the three origins. Although it becomes clear later in the manuscript (the NCCs are shared by two of them), I recommend to rephrase it so the NCCs are not confused with one of these three embryological origins.

Page 4, lines 79-80. Regarding the developmental origin of the limited morphological diversity of marsupials, it is controversial. In fact, one article cited in this sentence (Sanchez-Villagra 2013) argue against this hypothesis. I think that a caution note should be included and this citation highlighted as opposite evidence. In addition, there are more evidences against this hypothesis (although only for marsupial limbs: Martín-Serra and Benson 2020).

Page 8, lines 132-133. "The thylacine and wolf display more gradual shape change along the length of their trajectories". I do not see any obvious change in the slopes of these two species in comparison with the remaining ones in the figure S1. Maybe there are some results that support this claim and I missed, so I recommend to set this clearer. In any case, Could it be that the more gradual shape change is just the consequence of the greater size range of these two species in

comparison with the others?

Page 18, lines 320-322. The proxy for size used in the analyses of this paper is condylobasal length (CBL) but the widespread proxy for size in geometric morphometrics is centroid size (CS), obtained directly from the landmarks by the statistical package that the authors use. I think that the criteria the authors followed to make this choice (why CBL instead of CS) should be explained. In addition, CBL might be distorting a little bit the allometric results. The skulls of adult thylacines and wolfs are dolichocephalic in comparison with the young: setting size (CS or mesh volume) equal, they are longer. Therefore, CBL may overestimate size in the adults of these species and, hence, allometric effects may look greater. Most probably, this distortion would not change anything essential in the discussion and conclusions of the manuscript, but it would be good that readers could check that. It would be ideal that the authors include an analysis of allometry using CS in the supplementary material.

Reviewers' comments:

Reviewer #1 (Remarks to the Author):

General comments: This is a study about an interesting topic, the cranial convergence between mammals (a marsupial and a placental), approached with a suitable technique such as geometric morphometric. The structure is clear and it is well written. The graphics generated are very good. In general, materials and methods used are appropriate to the aim of the authors, although I found only one thing more complex to solve: Cranial similarities and/or differences are directly related to the species that we choose to compare. When a study includes species, which are morphologically extreme or have a very different diet (too small or with a completely different diet or prey size):

Bettongia penicillata (1.8 kg, Mycophagous), *Dasyurus viverrinus* (1.9 kg, eats insects and small vertebrates), *Phascolarctos cinereus* (11 kg, folivore), *Sminthopsis crassicaudata* (20 g, eats invertebrates), *S. macroura* (25 g, eats mostly invertebrates), *Trichosurus vulpecula* (4 kg, Mostly folivore)

these species gathered the greatest amount of variation, blurring more subtle patterns that may appear in the remaining species. In fact, this forces the similitude between *Thylacinus cynocephalus* and *Canis lupus* because they are more different. It would be more interesting perform the analysis with *Dasyurus maculatus* (1.5-5 kg, eats small vertebrates), *Sarcophilus harrissi* (8-14 kg, hypercarnivorous, eats medium vertebrates), and *Thylacinus cynocephalus* (29.5 kg, eats smaller prey relative to its size), and to compare them with *Canis lupus* (60 kg, hypercarnivorous). These three species belong to the carnivore guild of large mammals (Marsupials), and they are used in many works that report the convergence between *Thylacinus cynocephalus* and canids. If the authors change the sample by using three carnivorous medium-sized mammals (e.g. *Dasyurus maculatus*, *Sarcophilus harrissi*, and *Thylacinus cynocephalus*), I think that the pattern observed will be the same, but they will see the more subtle differences between *T. cynocephalus* and *Canis lupus* directly in a PC analysis without necessity to use cloud compare or other software to see that. I think that it would be a better comparison and reduce the analyses.

We thank the reviewer for this suggestion and agree that comparisons with larger marsupial carnivores would likely yield interesting results. Unfortunately, however, we are unable to accommodate this request due to the rarity and scarcity of postnatal specimens of these species in museum collections, which would be needed to include them in our analyses. To our knowledge, a digital developmental series does not exist for either the tiger quoll (*Dasyurus maculatus*; only the included *Dasyurus viverrinus*) or Tasmanian devil (*Sarcophilus harrissi*). Furthermore, tracking down and collecting pouch young and juvenile samples for CT scanning would be an exceedingly difficult and time-consuming task, where museum specimens often do not have age or stage associated with them (i.e. internally, or catalogued on iDigBio). Furthermore, these species are listed as threatened and endangered by the Australian government, with large efforts currently invested into breeding programs to increase viability in the latter. Therefore, obtaining new specimens of young individuals would not be feasible. In this study we attempted to sample as many developmental series for marsupials as we were able, though collecting these itself was challenging given their rarity (observed as the low number of individuals per species).

It should also be noted that in our first study of cranial shape variation among mammals (Feigin et al. 2018, Nat. Ecol. Evol. 2, 182–192), that we found no indication of cranial convergence between thylacine and *D. maculatus*, *Nimbacinus dicksoni* (a smaller, extinct thylacinid from the Miocene, but with a similar feeding ecology), or any of the other marsupials we included, at least for the adult morphology. There convergence was determined using three distance-based measures of phenotypic similarity relative to the ancestral phylomorphospace, that is, accounting for

phylogenetic relationships. In fact, of all the comparisons between adult crania, the only significant measures of convergent evolution were found between thylacine and species of *Canis* and *Vulpes*, with the strongest pairwise similarity to *Canis lupus*. While we realise that sampling cranial shape across postnatal ontogeny of these species could yield differing convergence patterns during development, the above findings give us confidence that the thylacine and *C. lupus* are in fact more similar to each other than the thylacine is to its own closest relatives, even considering those marsupials with comparable body sizes and ecologies.

Introduction

Line 43-45 “The thylacine and grey wolf (*Canis lupus*) are one of the most striking cases of convergent evolution in mammals, independently evolving nearly identical skull shapes ...” What about previous papers cited here that show more relationship between *Thylacinus cynocephalus* and *Vulpes vulpes* than *Canis lupus*?

Thank you for pointing out this error. These references were supposed to be after the previous sentence “the marsupial thylacine displayed remarkable similarities to placental canids” and are now updated in the text. The reference for line 43-45 should be (Feigin et al. 2018, Nature Ecology & Evolution) showing our previous comparisons of skull shape between the thylacine and canids. In this study we found that the thylacine and red fox (*Vulpes vulpes*) displayed the greatest similarity in cranial shape in the PC1 vs PC2 and PC1 vs PC3 morphospaces, but not between other components (as shown in Supplementary Figure 7 of Feigin et al. 2018), with the wolf displaying the next closest shape similarity). However, when we compared distances in ancestral shape space using scores from 31 PC axes, accounting for 99% of the total morphological variation, we found that the thylacine exhibited significantly greater overall similarity to the wolf than to the fox.

Line 73-75 “Placental mammals typically develop through an extended intrauterine gestation and are generally born in an advanced (precocial) state.” In the case of wolves, this is not accurate because they are altricial. Wolves are born deaf, blind and helpless. They are unable to move and even to regulate their own body temperature.

Thank you for pointing this out. Here we refer to the general precociality of eutherian mammals when compared to marsupial mammals. We acknowledge that this was not clear so have amended the text, lines 76-77. Also, we have added text that the newborn wolf is altricial, line 114-115, but not as altricial as the thylacine at birth, line 118.

The introduction would benefit from a brief paragraph where authors explain why is interesting the study of the comparative ontogeny. In addition, it would be a great moment to cite previous works in carnivore and marsupial ontogeny.

Thank you for this comment. We have integrated your suggestion into the introduction, lines 48-50.

Results

Line 112 Skull length 58.8 mm (mean)

The ZMZH specimen was not a newborn, so we considered only the four newborn MSU specimens, mean skull length 54.3mm

Line 114 more developed in which way? In relation to shape, to size? Please describe

More developed than the hyper-altricial thylacine.

Line 115 Skull length 37.1 mm (mean)

Only the 1.5 week old specimen was considered here.

Line 118 similar in which sense? Please describe.

Morphologically similar.

Line 111-120. Please modify the paragraph.

Thank you for the above comments. We have modified the paragraph to make our sampling clearer.

Line 126-127 What CAC means? Please explain this in material and methods section

We have replaced this graph with a scatter plot of individual cranial regression scores against logCBL, which is more appropriate when groups (here, species) have differing allometric trajectories, as we statistically show. This has been added to Fig 1 as a new panel, Fig 1c. Additionally, as per the comment regarding CBL vs centroid size, see answer below, we have included an additional supplementary figure showing a scatter plot of individual cranial regression scores against log centroid size (Fig S1).

Line 130-134 I would like a description about shape changes during ontogeny focused on the two species with phenotypic convergence instead of the other marsupials.

The homoplasious shape changes between the thylacine and wolf are detailed in the following paragraphs "Cranial disparity and convergence" and "Point-cloud visualization of cranial convergence". However, we have added additional text briefly describing the major observed shape similarities, lines 136-137.

Line 135 Please explain how the "Pairwise comparisons of slope vector lengths" was performed (in material and methods).

Text has been added to specify that pairwise comparisons were performed using the RRPP package in R to compare species least-squares means.

Line 150 In figure 2 the PC1 explained 54.4%

For simplicity we have listed whole numbers, so have rounded down to 54%. Updated in the text.

Discussion

Line 223 Please, try to integrate to the discussion the previous papers that compare the two same species.

To our knowledge, we have included all studies comparing cranial shape and convergence between the thylacine and wolf. If the reviewer has other specific papers in mind, we would be happy to consider them for citation.

Line 234-236 “These similarities are likely an adaptive response to the biomechanical demands of predation and shared ecological niches.”

The discussion would benefit from a deeper description of the relation to the cranial shape and the biomechanical demands and predatory behaviour that authors propose that the two compared species share.

Thank you for this suggestion. We have reworded these sentences and added an example of shared characteristics that may arise in response to their similar developmental trajectories.

Materials & Methods

Line 296-297 I cannot find digital cranial models neither in the paper nor in supplementary material for Ramírez-Chaves et al. 2016 (cited in Supplementary 1). Do materials acquire from published studies or from personal communication? Please modify if it is necessary.

The CT models for the marsupials utilized in the Ramírez-Chaves et al. 2016 study were provided to us by senior author of the study, and co-author of this paper, Dr Vera Weisbecker. All of these cranial models will be made available on MorphoSource with publication of this manuscript. In line with the comments of Reviewer 2, we have included a data availability statement to increase the transparency of the data used, and where they can be accessed.

Line 309-310 “...with age class determined by tooth eruption and life history data, when available.” Please, describe more the determination of age classes, maybe a table would be fine.

We have included an additional description of how we determined age class based off tooth eruption patterns, with associated references.

Line 315 Table S2 is not about landmarks.

Thank you for pointing out this error. Updated to Table S4.

Line 320-322 Why is necessary to take measures as CBL for comparisons if you have centroid size? You could perform regressions between log Centroid size and Procrustes coordinates.

CBL is a standard measure of skull length that is widely used in mammalian growth studies. In our experience, centroid size is generally used as a size proxy in many morphometric studies because they don't have access to actual size data. By measuring CBL for our included specimens, we put our findings in a real biological context where they can be directly related to organismal development, as centroid size is not an inherent property. However, as part of regular GM housekeeping, we tested the correlation between centroid size (CS) and CBL, which showed a strong linear relationship and was close to 1 ($r^2 = 0.9936$) and highly significant (<0.0001). We have added this information to the

text to make this clearer (line 328) and created a new supplementary figure (Fig S2). The regression of the Procrustes coordinates on CS also returned similar results to the ones we present using CBL, as would be expected given their near perfect correlation (Fig 1c vs Fig S1).

Line 346 more integrated.

Added to text.

Minor details

If it is possible, separate the figure 1 en two figures. The first of them should be figure 1B.

Thank you for the suggestion, in response to other comments we have altered figure 1 into three panels. As it stands now, Figure 1a is presented first in the text as an overview of our taxon sampling, Figure 1b shows landmark placement, and Figure 1c shows ontogenetic allometry.

Please add captions to supplementary material.

Added to end of manuscript.

Reviewer #2 (Remarks to the Author):

Newton and colleagues present a study of ontogenetic changes in the cranial morphology of the thylacine and the wolf, with additional analyses of those trajectories in the context of bone developmental and functional modules. They found broad convergence in the patterns of overall cranial shape changes between the two species, together with differences in the shifting modularity of within-cranium bone modules. This complex pattern of convergence and constraint is attributed in part to fundamental differences in the life histories of the two predators.

The manuscript is well-written, and the figures and tables are very informative for interpreting the authors' findings. This is a great study that adds clarity to the claim of morphological and ecological convergence between the thylacine and the wolf, through quantitative analyses of cranial shape changes through ontogeny. The statistical methods appear to be robust and carefully chosen. The comparative nature of such morphological analyses typically would require phylogenetic comparative methods to account for phylogenetic statistical non-independence; in the case of this study, the divergence times between thylacine and wolf are so deep, any effect of phylogenetic relatedness in the metatherian part of the dataset most likely would not have altered the conclusions presented by the authors. The figures add greatly to the interesting results reported.

We thank the reviewer for these kind comments.

I have just two suggestions for improvement, both concerning data reporting:

-P17L310: according to Table S1, the CT images used in this study are a combination of openly accessible and not openly accessible data. For maximum transparency, please provide explanations

for the different accessibility levels of the raw data used in the analyses. Neonatal and juvenile specimens of any wild mammal species are uncommon in museum collections, and especially so for an extinct taxon such as the thylacine. The importance of this dataset cannot be overstated, for future work that build on the current study, as well as other lines of research on the ontogeny and evolution of the thylacine. Therefore, increasing data accessibility could be one of the significant contributions presented by this study.

Thank you for this suggestion. We advocate for open data and will make all of our generated CT data, and unpublished cranial models we utilized, publicly available. In accordance, we will upload these in the MorphoSource project P1124, which has been included to the data accessibility statement at the end of the manuscript.

-P20L370: surely the analyses were done with the usual ethical standards in mind (Data free of manipulation, digital datasets used with permission, etc.)? I suggest "No special ethical considerations were necessary in this study" instead.

Updated in the text.

Reviewer #3 (Remarks to the Author):

The manuscript entitled “Ontogenetic origins of cranial convergence between the extinct marsupial thylacine and placental grey wolf” aim to find similarities and differences in the ontogenetic growth patterns between the thylacine and the wolf, two species that represent one of the most iconic examples of convergence between marsupials and placentals. Their starting hypothesis is that some similarities should be found, but also some differences due to the characteristic developmental constraints of each group. They also explore patterns of modularity and integration through ontogeny, which are very important aspects that influence morphological change. In my opinion, the hypotheses are clear, the analyses are correct, and their conclusions are supported by the results. I strongly recommend the publication of this article in *Communications Biology*. I just have some suggestions for improving the manuscript that the authors may find useful. They are detailed below:

Page 3, line 45. There are also evidences of a lack of convergence in their predatory behavior or hunting strategy, as shown by Figueirido and Janis (2011) and Janis and Figueirido (2014).

Thank you for pointing this out. We are aware of these studies showing distinct forelimb anatomies between the thylacine and wolf, and have included these into the introduction (line 45). However it's important to note that while these differences exists, we are solely examining their similarities in cranial shape. Additionally, we have included these references into our discussion line 237.

Page 3, line 52. In my opinion, this sentence is not clear enough about which are the three origins. Although it becomes clear later in the manuscript (the NCCs are shared by two of them), I recommend to rephrase it so the NCCs are not confused with one of these three embryological origins.

Excellent suggestion, we have reworked the text to make this distinction clearer.

Page 4, lines 79-80. Regarding the developmental origin of the limited morphological diversity of marsupials, it is controversial. In fact, one article cited in this sentence (Sanchez-Villagra 2013) argue against this hypothesis. I think that a caution note should be included and this citation highlighted as opposite evidence. In addition, there are more evidences against this hypothesis (although only for marsupial limbs: Martín-Serra and Benson 2020).

Thank you for pointing out this controversy. We have included additional discussion and references outlining the opposing evidence towards this hypothesis, lines 84-86.

Page 8, lines 132-133. “The thylacine and wolf display more gradual shape change along the length of their trajectories”. I do not see any obvious change in the slopes of these two species in comparison with the remaining ones in the figure S1. Maybe there are some results that support this claim and I missed, so I recommend to set this clearer. In any case, could it be that the more gradual shape change is just the consequence of the greater size range of these two species in comparison with the others?

In the now updated Fig. 1c showing individual cranial regression scores against CBL (and Fig S1 showing regression vs centroid size), both the thylacine and wolf cover a smaller range of cranial shape change (as seen in the range of regression scores) over their postnatal ontogeny than do any

other species except *Phascolarctus*. However, this is minor so included the word “slightly” in the text, line 134.

Page 18, lines 320-322. The proxy for size used in the analyses of this paper is condylobasal length (CBL) but the widespread proxy for size in geometric morphometrics is centroid size (CS), obtained directly from the landmarks by the statistical package that the authors use. I think that the criteria the authors followed to make this choice (why CBL instead of CS) should be explained. In addition, CBL might be distorting a little bit the allometric results. The skulls of adult thylacines and wolves are dolichocephalic in comparison with the young: setting size (CS or mesh volume) equal, they are longer. Therefore, CBL may overestimate size in the adults of these species and, hence, allometric effects may look greater. Most probably, this distortion would not change anything essential in the discussion and conclusions of the manuscript, but it would be good that readers could check that. It would be ideal that the authors include an analysis of allometry using CS in the supplementary material.

Thank you for this comment, this is an interesting point. As per our response to Reviewer 1, we chose to use CBL as it is a standardised measure of skull length which can be directly used and compared with other studies (as centroid size is not an inherent property of skulls unless researchers perform the same landmark analysis). Also, we did not find any consequence of dolichocephalic scaling of thylacine and wolf cranial shape with our CBL measurements. This was observed where CBL and centroid size exhibited a strong linear relationship and neither the thylacine nor wolf deviated from the regression using CBL measurements. These similarities are seen in regression score vs CBL (Fig 1c) and centroid size (Fig S1), and clearly observed in CBL vs CS in Fig S2. We have additionally added text to the methods (line 327-328) reporting the correlation of CBL with centroid size.

Reviewers' Comments:

Reviewer #1:

Remarks to the Author:

I reviewed the previous version of this manuscript and I think the authors did a lot of work to improve the manuscript. The result is noticeably better.

However, I have to insist about the sample.

The study includes species morphologically extreme and have different diet compared to *Thylacinus cynocephalus*, which is a carnivorous that eats smaller prey and weighs 29.5 kg.

The authors used for comparison

Bettongia penicillata (1.8 kg, Mycophagous),

Dasyurus viverrinus (1.9 kg, eats insects and small vertebrates),

Phascolarctos cinereus (11 kg, folivore),

Sminthopsis crassicaudata (20 g, eats invertebrates),

S. macroura (25 g, eats mostly invertebrates),

Trichosurus vulpecula (4kg, Mostly folivore)

They are inadequate because they could gather the greatest amount of variation, blurring more subtle patterns that may appear in the remaining species.

Sarcophilus harrissi is a better candidate for this comparison because it is a hypercarnivorous, eats medium vertebrates and weighs 8-14 kg.

No need to collect new material in the field. In mammal collections, for example, Australian Museum, American Museum, Smithsonian Museum, *Sarcophilus harrissi* is a common species. The authors can obtain postnatal ontogenetic series there. The authors do not need to have CT scans of this species; they can digitize the material with a Microscribe or taking photographs and then put the same landmarks, and combine the sources.

If the authors modify that, I think that the pattern observed will be the same, but they will see the more subtle differences between *T. cynocephalus* and *Canis lupus* directly in a PC analysis. Therefore, I would like to encourage the authors to improve the sample.

REVIEWERS' COMMENTS:

Reviewer #1 (Remarks to the Author):

I reviewed the previous version of this manuscript and I think the authors did a lot of work to improve the manuscript. The result is noticeably better. However, I have to insist about the sample. The study includes species morphologically extreme and have different diet compared to *Thylacinus cynocephalus*, which is a carnivorous that eats smaller prey and weighs 29.5 kg. The authors used for comparison:

- *Bettongia penicillata* (1.8 kg, Mycophagous),
- *Dasyurus viverrinus* (1.9 kg, eats insects and small vertebrates),
- *Phascolarctos cinereus* (11 kg, folivore),
- *Sminthopsis crassicaudata* (20 g, eats invertebrates),
- *S. macroura* (25 g, eats mostly invertebrates),
- *Trichosurus vulpecula* (4kg, Mostly folivore)

They are inadequate because they could gather the greatest amount of variation, blurring more subtle patterns that may appear in the remaining species.

Sarcophilus harrissi is a better candidate for this comparison because it is a hypercarnivorous, eats medium vertebrates and weighs 8-14 kg.

No need to collect new material in the field. In mammal collections, for example, Australian Museum, American Museum, Smithsonian Museum, *Sarcophilus harrissi* is a common species. The authors can obtain postnatal ontogenetic series there. The authors do not need to have CT scans of this species; they can digitize the material with a Microscribe or taking photographs and then put the same landmarks, and combine the sources. If the authors modify that, I think that the pattern observed will be the same, but they will see the more subtle differences between *T. cynocephalus* and *Canis lupus* directly in a PC analysis. Therefore, I would like to encourage the authors to improve the sample.

Our previous analysis in *Feigin et al. (2018)* examining cranial shape disparity between marsupials and carnivora (**Figure 3**) found that the thylacine in fact displays greater general cranial shape similarity to smaller-bodied Dasyurids (i.e. *Antechinus swainsonii*, *Phascogale tapoatafa*, *Dasyurus viverrinus* and *Dasyurus geoffroi*) and the numbat (*Myrmecobius fasciatus*) than it does with the Tasmanian Devil (*Sarcophilus harrissi*) and tiger quoll (*Dasyurus maculatus*). This was largely due to the “long, narrow faced” morphology of the thylacine and dasyurids, compared to the devils and quolls more derived “short, wide faced” morphology. While we are aware that this additional comparison may yield some interesting findings due to greater similarities in body mass, we believe that in this instance, our current comparisons represent greater similarity in “shape”. Perhaps instead the addition of other dasyurids (i.e. *Phascogale*) may reveal these subtle differences, though as supported by the editor, finding ontogenetic series for these rare taxa would be difficult in this current climate.

It is important to acknowledge however that this is a substantial limitation of this study and perhaps something that should be scrutinized in greater detail in subsequent analyses. As such, we have included additional insights regarding these limitations into the methods, and first paragraph of the discussion, lines 232 – 235.